# Investigation of Synergistic Effects and Kinetics on Co-Pyrolysis of *Alternanthera philoxeroides* and Waste Tires

**DOI:** 10.3390/ijerph19127101

**Published:** 2022-06-09

**Authors:** Awsan Shujaa Aldeen, Jiapeng Wang, Bo Zhang, Shuying Tian, Zhixiang Xu, Huiyan Zhang

**Affiliations:** 1Key Laboratory of Energy Thermal Conversion and Control of Ministry of Education, School of Energy and Environment, Southeast University, Nanjing 210096, China; aosanshugaa@outlook.com (A.S.A.); 220210588@seu.edu.cn (J.W.); 213171445@seu.edu.cn (S.T.); 220210489@seu.edu.cn (H.Z.); 2School of Energy and Power Engineering, Jiangsu University, Zhenjiang 212013, China; xzxjsu@163.com

**Keywords:** co-pyrolysis, *A. philoxeroides*, waste tires, catalyst

## Abstract

A thermogravimetric analysis is used to analyze the thermal kinetics and investigate the synergistic effects between *Alternanthera philoxeroides* (AP) and waste tires (WTS) in a temperature range of 50–900 °C under three heating rates (15, 25, and 35 °C/min). Two model-free methods (FWO and KAS) and a model-fitting method (CR) were applied to calculate the activation energy. Results revealed that heating rates had no significant effect on the pyrolysis operation. The addition of WTS improved the thermal degradation of the samples as the samples had more than one stage during the main reaction period. A promoting synergistic effect was found in the blend 75A25WT and obtained the lowest activation energy among all the blends without a catalyst, while the blend 50A50WT exhibited an inhibiting effect. On the other hand, the addition of HZSM-5 accelerated the reaction time and obtained the lowest activation energy among all the blends without a catalyst. Furthermore, ΔW of 75A25WT+C was the lowest, indicating that the blend with a catalyst exhibited the strongest synergistic effect. This research confirmed that the addition of WTS improved the thermal parameters of the samples and clarified the capacity of HZSM-5 to reduce the activation energy.

## 1. Introduction

Resource depletion and environmental problems are becoming increasingly severe as a result of the excessive consumption of fossil fuels, which has become a global concern in recent years. Nowadays, there has been a significant increase in scientists looking at clean and sustainable energy alternatives in response to this issue. Biomass is considered an essential potential fuel source among these energy sources since it is a renewable energy source created by photosynthesis. Biomass has historically been a significant source of energy for humanity and is currently projected to generate between 10% and 14% of the world’s energy supply. With the use of several technologies, such as rapid pyrolysis, enzymatic hydrolysis, liquefaction, gasification, and other processes, biomass can be converted into chemicals and fuels, making it an excellent substitute for fossil fuels [1,2,3].

*Alternanthera philoxeroides* is an Amaranthaceae weed from South America. These invasive weeds may have arrived in India with some packaging debris during WWII. The species was discovered in India near a Calcutta airfield [4]. *Alternanthera philoxeroides* is a perennial weed that is aquatic, versatile, and extensively dispersed. It proliferates and has a considerable impact on agricultural productivity. Alligator weeds diminish oxygen content and light penetration in water, reducing water flow. It also harms zoological and botanical habitats, killing fish, birds, and other aquatic plants. It is tough to control and aids in mosquito breeding. *A. philoxeroides* releases toxins throughout its decline stage, lowering the biological oxygen demand of submerged bodies.

Previous research has demonstrated that these invasive wetland plants can be used as bio-oil, animal feed, wastewater treatment, and pyrolysis feedstock [5]. The aquatic plant harvester can be used to gather and transport these invasive species to processing plants located nearby. As a result, *A. philoxeroides* may become a source of diverse biomass for energy conversion processes.

Thermochemical processes such as combustion, torrefaction, hydrothermal liquefaction, gasification, and pyrolysis are considered the fastest way to convert biomass into valuable products. Fast pyrolysis of biomass has garnered considerable attention among scientists due to its enormous potential for converting biomass into fuel [6]. It requires 450–650 °C reaction temperature, 103–104 K/S heating rate, and <2 S residence time. Biomass is broken down into short-chain molecules and then condensed into bio-oil throughout this process [7]. However, the raw bio-oil generated is still not immediately applied for industrial production and daily life due to its high oxygen contents, low heating value, low thermal stability, and corrosiveness [8,9]. Thus, adding catalysts can increase the yield and the quality of bio-oil products produced by the pyrolysis process and reduce the activation energy. Previous research has shown that both CaO and zeolite catalysts have outstanding catalytic performance in biomass pyrolysis [10,11]. For example, zeolite catalysts could reduce the amount of oxygen compounds and other undesirable chemicals. During the regeneration cycles, a significant amount of aromatic content was observed in the bio-oil, which significantly enhanced bio-oil stability [10]. Aside from producing valuable products, the CFP conversion technique has many drawbacks, such as coke formation, catalysts deposition, blockage of micropore, catalysts deactivation, and production of products with high (PAHs), which limit their use as PAHs are carcinogenic and harmful to the environment [12]. Furthermore, biomass has low hydrogen to carbon effective ratio, H/Ceff, and it produces a significant amount of coke while yielding a low amount of aromatic chemicals [13]. Many recent research studies have concluded that coke formation is the main impediment to the developing of catalyst-fast pyrolysis [14]. Hence, co-pyrolysis of biomass and plastic waste can provide an excellent solution due to its great potential to improve the yield and the quality of bio-oil as well as limit coke formation [15].

Co-pyrolysis of biomass and polymer materials wastes has garnered considerable attention among scientists in recent years because of its enormous potential to replace fossil fuels and address the world’s ever-growing municipal solid waste problem [16]. Among these polymers’ materials, waste tires contain a high concentration of hydrogen and volatiles, while oxygen concentrations are low. These features make the waste tires suitable for co-pyrolysis with biomass as a hydrogen source. Thus, adding plastics to biomass pyrolysis may increase the H/Ceff ratio [17,18,19]. Studies have also shown that co-pyrolysis of biomass and plastic yields high-quality bio-oil with a higher calorific value [20]. During co-pyrolysis, the contact between the biomass and waste tires causes a synergistic effect [21]. The radicals released by the breakdown of biomass start the depolymerization of waste tires. Subsequently, polyolefin breakdown products generate radical–biochar interactions to form 2-alkanes. The hydrogen released from plastic accelerates cellulose decomposition in biomass, and the oxygenated compounds accelerate plastic breakdown [15,22,23].

Earlier reports studied the influence of different catalysts on product yields, product characterizations, and changes in *Alternanthera philoxeroides* fuel characteristics. It was found that catalysts reduced oxygen content and enhanced biochar’s heating value [24,25]. Wang et al. detected a synergistic effect between *Alternanthera philoxeroides* and peanut soapstock, which facilitated aromatics production in the bio-oil. Moreover, the results revealed that increasing the quantity of catalysts might improve bio-oil quality while decreasing its yield [26]. However, no such study highlights the effect of the mixture ratio of AP, WTS, and catalyst on the reaction mechanism. It is necessary to investigate in which condition co-pyrolysis of AP could achieve the most significant synergistic effects to further enhance the quality derived from bio-oil and the efficiency of the bio-energy utilization.

This study conducted co-pyrolysis of a mixture of *A. philoxeroides* and waste tires in a continuous fast microwave catalytic system. A thermogravimetric analyzer (TGA) is used to investigate the impact of the mixture ratio and catalyst on thermal cracking. As a part of this work, the kinetic analysis of *A. philoxeroides* and waste tires and their mixtures was conducted. The activation energy was calculated using model-free methods (FWO and KAS) and a model-fitting method (Coats–Redfern method).

## 2. Experimental Method and Equipment

### 2.1. Materials

*A. philoxeroides* were collected from a lake in Nanjing. Waste tires were purchased from Nanjing Honghua Tire Company in Nanjing city, Jiangsu Province, China. *A. philoxeroides* (AP) and waste tires (WTS) were pulverized into powders that passed through 40 and 100 mesh sieves. An elemental analyzer (Euro Vector EA3000, Italy) was used for the ultimate analysis of the samples. On the basis of Chinese National Standards (GB/T 28731, 2012), a proximate analysis was performed. Table 1 shows the analysis of dried *A. philoxeroides*. To minimize the water content in the AP and WTS powders, they were dried in an oven for 12 h at 105 °C. The samples were mixed in different ratios of 0:1, 1:3, 1:1, 3:1, and 1:0 (expressed as WT, 25A75WT, 50A50WT, 75A25WT, and A). HZSM-5 zeolite catalyst utilized in this work has a silicon-to-aluminum ratio of 38 and a particle diameter of 0.53–0.58 nm, which was purchased from Nanjing Catalyst Factory. Prior to the experiment, the HZSM-5 was activated by calcining it for 5 h at 550 °C in a muffle furnace. The sample containing catalyst is expressed as 75A25WT+C.

### 2.2. Thermogravimetric Experiment

In this work, co-pyrolysis of *Alternanthera philoxeroides* (AP) and waste tires (WTS) was examined using a thermogravimetric analyzer (NETZSCH 449 F3, Germany) to evaluate the pyrolytic characteristics by obtaining the mass loss (TG) curves and the mass loss rate (DTG) curves. The mixtures were heated to 900 °C under three heating rates of (15, 25, and 35 °C/min). The nitrogen gas flow rate was (40 mL/min). The weights of the samples were 5 mg, and the mixes’ ratios were (0:1, 1:3,1:1, 3:1, 1:0). In total, 5 mg of HZSM-5 was added to 75A25WT+C.

### 2.3. Kinetic Parameter Analysis

Thermal degradation of biomass entails a number of complicated reactions involving a variety of chemical elements that occur concurrently. This makes it difficult for adopting basic kinetic models. For solid materials, the TG/DTG technique is highly effective for determining decomposition temperature and kinetic parameters. Thermal degradation processes’ kinetics are characterized using a variety of equations that take into consideration the unique characteristics of their mechanisms. Iso-conversional and model-fitting kinetic analysis techniques have been categorized as two distinct approaches to the study of kinetics [27,28]. This study employed the model-fitting method (CR) and model-free methods (Flynn–Wall–Ozawa (FWO), Kissinger–Akahira–Sunose (KAS)) for the kinetic analysis of AP, WTS, and their mixtures.

The kinetic equation for the solid-state decomposition rate and rate of conversion can be described by the Arrhenius equation:(1)$$\frac{da}{dt}=k\left(T\right)f\left(a\right)=Aexp\left(-\frac{E}{RT}\right)f\left(a\right)$$
where A, E, T, R represent the pre-exponential factor, the apparent activation energy of the reaction, reaction temperature (K), and the gas constant (R = 8.314 J ${\mathrm{mol}}^{-1}K^{-1}$). $f\left(a\right)$ represents the reaction model and a is the conversion degree which can be obtained from the equation below:(2)$$a=\frac{w_{0}-w_{t}}{w_{0}-w_{f}}$$
where $w_{0}$, $w_{f}$ are the initial and final weight of the sample and $w_{t}$ represents the weight of the sample at a time. By defining $\beta =\frac{da}{dt}$, we can rewrite the equations above as follows:(3)$$\beta =\frac{da}{dt}=Aexp\left(-\frac{E}{RT}\right)f\left(a\right)$$

By integrating the equation, we get the following equation:(4)$$G\left(a\right)=\overset{a}{\underset{0}{\int }}\frac{da}{f\left(a\right)}=\frac{A}{\beta }\overset{T}{\underset{0}{\int }}e^{-E/RT}dT$$

#### 2.3.1. Model-Free Methods

Two iso-conversional models, Kissinger–Akahira–Sunose (KAS) and Flynn–Wall–Ozawa (FWO), were used to obtain the activation energy. (KAS) and (FWO) are shown in the equations below, respectively:(5)$$ln\left(\frac{\beta }{T^{2}}\right)=ln\frac{AR}{EG\left(a\right)}-\frac{E}{RT}$$
(6)$$ln\beta =ln\frac{AR}{RG\left(a\right)}-5.3305-1.052\frac{E}{RT}$$

In both approaches, the activation energy is dependent on the conversion rate. After selecting a particular value of a, the activation energy can be obtained from the slope by plotting $ln\left(\frac{\beta }{T^{2}}\right)$ and lnβ verses 1/T [29,30].

#### 2.3.2. Model-Fitting Method (Coats–Redfern (CR))

CR is an integral method used to determine the activation energy E and the pre-exponential factor A [31], which is shown in the following equation:(7)$$ln\left[G\left(a\right)\right]=ln\frac{AR}{\beta E}-\frac{E}{RT}$$
where $G\left(a\right)$ is the function for the different reaction mechanisms, in this work, the reaction order is selected to be 1, as shown in the equation below:(8)$$G\left(a\right)=\frac{-ln\left(1-a\right)}{T^{2}}$$

By using the data of a and T in the TG curve, we can determine the value of E by obtaining the slope and intercept of ($\frac{-ln\left(1-a\right)}{T^{2}})$ vs. 1/T with a specific heating rate β [32].

## 3. Results and Discussion

### 3.1. Effect of Heating Rate on Pyrolysis Process

As an aquatic plant, *A. philoxeroides* has a high water content (higher than 90%), and the main components of dry matter are cellulose, hemicellulose, and lignin (about 33%–55%) [26]. The TG and DTG curves of *A. philoxeroides*, waste tires, and their blends’ pyrolysis at different heating rates are shown in Figure 1 and Figure 2.

It can be seen that the pyrolysis process can be divided into four stages [33]: (1) the drying and dehydration stage (50–150 °C), where the water in the sample evaporates; (2) the devolatilization stage (130–260 °C) where the volatile matter in the sample is removed, and the hemicellulose in the sample plant is thermally decomposed; (3) the combustion stage (250–430 °C) where the cellulose and lignin with larger molecular weight in the sample are thermally decomposed, and the reaction rate decreases with the increase in temperature; and (4) the residual calcination stage (≥400 °C) where the lignin in the sample undergoes subsequent calcination. This sequence of pyrolysis stages is consistent with the conclusion of [34], which states that hemicellulose begins to decompose around 200–350 °C, and cellulose begins to decompose around 350–400 °C. In the case of WTS, one decomposition step was observed, with a sharp peak at 379.5 °C, which is attributed to the decomposition of natural rubber (polyisoprene, NR) [35].

The TG and DTG curves of AP and WTS with and without HZSM-5 catalyst show that different heating rates do not affect the whole pyrolysis process. The pyrolysis shows similar peaks at similar time temperatures. It can be found from the DTG curve (Figure 1 and Figure 2) that with the increase in heating rate, the maximum loss rate tends to move to a higher temperature. The reason may be that the heat transfer efficiency in the sample is low at a high heating rate, which cannot provide enough heat to heat the surface and interior of the heated object at the same time [36]. When the heating rate is 15 °C/min, the weight loss after pyrolysis is the lowest in most cases. The possible reason is that hemicellulose and cellulose are not pyrolyzed completely, and less volatiles were released [37]. Although the highest weight loss rate is often found when the heating rate is 35 °C/min, the peak of the DTG curve masks some subtle changes at a high temperature and cannot be well analyzed for specific peaks [38]. Therefore, the data with a heating rate of 25 °C/min are selected to further analyze the pyrolysis process.

### 3.2. The Effect of Sample Mixing Ratio and Catalyst on the Pyrolysis Process

The TG and DTG curves of AP and WTS at a heating rate of 25 °C/min are shown in Figure 3. The graph shows that AP starts to decompose at a low temperature of 50 °C because of the existence of water content in the AP. While WTS thermal decomposition occurs at high temperatures due to the molecular nature of plastic, it requires a greater temperature to break its chemical bond. For each of the two feedstocks, the raw material had a single peak in entirely distinct temperatures. Pyrolysis of most different types of biomasses is divided into three simple phases [39]. The first phase in AP pyrolysis occurred between 50 °C and 150 °C, where there was a 4.45% loss rate due to the existence of moisture and small volatiles. As an aquatic plant, the second phase, which is the main phase, has three stages.

Table 2 shows the parameters of the three stages sorted from TG and DTG curves. The first stage happened between 150 °C and 244 °C, resulting in a loss rate of 8.5% owing to hemicellulose degradation and generation of volatiles (carbon dioxide and carbon mono-oxide) as well as loss of oxygenation, nitrogen, and other elemental components. The second stage happened between 250 °C and 356 °C, resulting in a maximum loss rate of 9.56% due to cellulose decomposition. The third stage occurred between 369 °C and 530 °C due to lignin and other solid residue decomposition. The final weight loss was 30% at 800c. However, the second decomposition phase of WTS only had one stage ranging from 290 °C to 510 °C with a maximum loss rate of 15.278% at 380 °C. TG graph shows that the weight loss of WTS remained unchanged after 510 °C resulting in a final weight of 37%, which is attributed to the existence of impurities in WTS that are unable to decompose. It has been reported that waste tires contain various components such as (NR, BR, and SBR) [20].

Figure 4 shows the TG and DTG curves for different blends. The co-pyrolysis of the blends resulted in more complicated thermal behavior than the pyrolysis of AP and WTS alone. A significant change happened in the pyrolysis of the WTS when mixed with AP, which could indicate a synergistic effect between the two mixtures. From the figure above, we notice that the mass loss curves of the blends move to lower temperatures as the biomass ratio increases. When the co-pyrolysis of AP and WTS mixtures happened, the $T_{i}$ of AP increases while the $T_{i}$ of WTS decreases. This phenomenon is due to the melted WTS covering the surface of the AP, preventing the volatile from releasing [40]. Moreover, the degradation products of AP can accelerate the decomposition of WTS due to the free radicals generated during the devolatilization of AP, which attack the molecules of WTS, causing a drop-in $T_{i}$ of WTS [41]. The blends’ degradation showed a similar trend with three stages in the second phase of the reaction, except for 25A75WT. It was apparent that the melted WTS hindered the decomposition of AP during the first stage. With the increase in temperature, WTS starts to decompose, resulting in a maximum loss rate of 14% at 385 °C among all the blends. This indicates that the free radicals generated from AP enhanced the reaction [41]. The existence of a catalyst in the blend shifted the initial and the final reaction temperature in the second phase to a lower temperature, which indicates that HZSM-5 improved the reaction efficiency by lowering the activation energy [42]. Moreover, the final weight loss in 75A25WT+C was the lowest, with a final solid residue of 29%, implying that HZSM-5 could improve the pyrolysis process and facilitate the reaction.

### 3.3. The Analysis of the Synergistic Effect of the Blends

A synergistic effect calculation becomes essential when two distinct materials are combined and pyrolyzed together in a single reaction which deals with the positive and negative interactions during the operation. In order to detect the synergistic effect, the interaction between the weight loss obtained from the experiment and the theoretical weight loss needs to be calculated. The interaction was computed using the additive formula as shown in the following equations:(9)$$W_{cal}=x_{1}W_{A}+x_{2}W_{WT}$$
(10)$$\Delta W=W_{exp}-W_{cal}$$
where $x_{A}$ and $x_{wt}$ represent the percentage of AP and WTS in the mixture, respectively, while $W_{A}$ and $W_{WT}$ are the weight losses of Ap and WTS. $W_{exp}$ is the weight loss obtained from the TG curve. Figure 5 shows the fluctuation in $W_{exp}$ and $W_{cal}$ values of the blends with and without a catalyst at a heating rate of 25 °C/min.

Quan et al. claim that an increase or reduction in ΔW indicated a synergistic impact [43]. Researchers defined the degree of interaction as follows: negative ΔW values demonstrate a promoting effect, whereas positive ΔW values demonstrate an inhibitory effect [44]. From the figure above, it can be seen that the fluctuation of ΔW is near zero for all the blends until 200 °C; during this period, neither Ap nor WTS began to degrade. Moreover, it indicates that the interactions between AP and WTS have not happened. Furthermore, the starting weight of each component varies; hence the ΔW is not zero at this point. This indicates that the interactions occur at high temperatures. The ΔW of all the blends without a catalyst started to fluctuate at around 210 °C. However, the ΔW of the blend 50A50WT was the closest to zero, which could be attributed to the reduction in WTS proportion in the blend, reducing the amount of covered biomass. For the blend 25A75WT, ΔW started to increase at around 210 °C, peaking at 360 °C and then rapidly dropping from a positive to a negative value, reaching a minimum value at 490 °C. This fluctuation can be described by the physical status of WTS. First, it melts and prevents the volatiles from releasing by forming a coat on the surface of AP. With the increase in temperature, the internal pressure of volatiles increases and breaks through the coating layer. It increases the rate of pyrolysis of WTS, causing the decline in ΔW. This indicates that WTS aided the inhibitory impact.

The ΔW of the blend 75A25WT was always less than zero, indicating a synergistic effect during co-pyrolysis. It can be seen that several peaks occurred during the reaction. The first peak happened at 246 °C due to the decomposition of AP. The second and third peaks occurred at 360 °C and 395 °C due to the free radicals generated from AP devolatilization, causing the degradation of WTS and the release of hydrocarbon radicals [41,45]. All the blends without a catalyst completed the pyrolysis at 490 °C. Regarding the blend 75A25WT+C, it can be seen that ΔW started to decline at a very low temperature which indicates that the catalyst helped initiate the reaction between AP and WTS. The blend also had several peaks due to the decomposition of hemicellulose, cellulose, and lignin, the free radicals generated during the process attack the WTS molecules causing ΔW to increase. The blend 75A25WT+C exhibited a maximum negative synergistic effect at 205 °C, among all the blends, indicating the catalyst’s ability to improve the interactions between AP and WTS. The divergence toward a positive direction might be attributed to char’s adsorption of volatile chemicals due to secondary reactions. Based on the prior process, it was evident that the blends exhibited a degree of interaction; the blend 25A75WT and 50A50WT exhibited an inhibiting effect, while the blends 75A25WT and 75A25WT+C exhibited a promoting effect.

### 3.4. Kinetic Analysis

The activation energy measures the minimal amount of energy required to initiate a thermochemical process. Materials which have low activation energy are easier to pyrolyze and faster to start reactions [46]. Due to the intricacy of the co-pyrolysis process, two iso-conversion techniques were utilized to estimate the activation energy to ensure the precision of the calculated results [47].

#### 3.4.1. Model-Free Method

Using TG data at three heating rates (15, 25 and 35 °C/min), the Kissinger–Akahira–Sunose (KAS) and Flynn–Wall–Ozawa models were used to calculate AP, WTS activation energy, and their blends with and without HZSM-5. Table 3 shows the estimated activation energy (E) and the correlation coefficients ($R^{2}$). The blends had correlation coefficients between 0.935 and 0.999 when the conversion degree ranged between 0.1 and 0.9, demonstrating the accuracy of the models employed in the calculation. Furthermore, there is no significant difference between the values obtained from FWO and KAS, which indicates that the values are valid. A comparison of activation energy and conversion degree distributions for AP, WTS, and their blends with HZSM-5 is shown in Figure 6.

Table 3 shows that the activation energy of AP ranges from 34.351 KJ/mol to 307.210, with an average of 147.356 and 147.204 KJ/mol. It was observed that the activation energy of the AP increased from α=0.5 to α=0.8 and exceeded the value of WTS. This could attribute to the lignin calcination stage of biomass, which occurred at a temperature range of 320 °C–430 °C. Furthermore, the weight loss rate is low, making the activation energy required for the reaction larger. The fluctuations within the conversion degrees ranging between 0.7 and 0.9. It might be related to the degradation process behaviors of the AP biomass as the weaker bonds commonly degraded at comparatively lower temperatures, while the stronger bonds took more energy to fracture at higher temperatures.

For waste tires, the activation energy ranges from 80.929 to 215.085 KJ/mol, with an average of 142.161 and 145.571 KJ/mol, which is less than the AP’s activation energy, indicating the poor thermal stability of waste tires.

The E of the blend 25A75WT is the highest, ranging from 32.023 to 394.179 KJ/mol, with an average of 214.946 and 221.772 KJ/mol. The activation energy increased from 0.1, reaching its maximum at 0.3 with an activation energy of 394.2 KJ/mol. The reason here is that the inhibition of WTS on biomass decomposition during the melting process will increase the activation energy. This implies that a low biomass feedstock proportion may affect the performance of co-pyrolysis. In comparison, the activation energy of 50A50WT had a similar trend to the WTS and obtained an intermediate value in the range of 31.028 to 196.032 KJ/mol with an average of 96.233 and 93.788 KJ/mol. The blend 75A25WT obtained the lowest activation energy compared to other blends, with an average of 73.458 and 70.641 KJ/mol. The energy was constantly at a bare minimum, and only minor changes could be seen.

Figure 7 shows the comparison between the experimental and the calculated activation energy ($E_{C})$ from the equation below:(11)$$E_{C}=x_{A}E_{A}+x_{wt}E_{wt}$$
where $x_{A}$ and $x_{wt}$ are the mass ratios of AP and waste tires in the mixture. $E_{A}$ and $E_{wt}$ are the average activation energies of AP and WTS obtained by FWO and KAS methods. Noticeable differences in $E_{C}$ for the blend’s ratio were obtained and shown in Table 4.

These variances are due to differences in the major component and mixture structures. As can be observed, when the proportion of AP is 50% and above, the experimental result was lower than the estimated value, showing that the proper mixing ratio increased the activity and decomposability of the blends. The discrepancy between the components’ activation energies and their estimated activation energies suggests the presence of synergistic effects. This effect is attributed to the chemical interactions between evolving volatile molecules, which promote the volatile matter amount from the components. On the other hand, separated volatile could result in higher activation energy due to the surface’s transferer energy or mass limitation.

In the case of 75A25WT+C, the activation energy ranges from 1.166 to 93.650 KJ/mol with an average of 53.076 and 49.404 KJ/mol, which is lower than all activation energies obtained from the blends with no HZSM-5 catalyst. It indicates that using HZSM-5 can decrease the activation energy without affecting the pyrolysis mechanism.

#### 3.4.2. Model-Fitting Method (Coats–Redfern)

The CR approach, which is frequently used in non-isothermal kinetic research, was utilized in this investigation to estimate the kinetics of AP, WTS, and their blends at a heating rate of 25 °C/min and a first-order reaction. The slopes and intercepts of the samples used to determine the kinetic parameters are shown in Figure 8. The values of the activation energy and the correlation coefficient $R^{2}$ are listed in Table 5.

The correlation coefficient $R^{2}$ was determined to be (>0.97), which showed the best fitting curve. Compared to the results acquired from the model-free methods, the activation energy values obtained from CR were lower than those obtained from the model-free methods due to the different algorithm approximations. Additionally, the CR model analyzes the main pyrolysis peak, and the temperature involved is smaller than that of the model-free methods. The other two methods involved some regions with higher activation energy, which will make the overall activation energy higher. However, 75B25H had the lowest Ea (3.982 kJ/mol), followed by 50A50WT (7.108 kJ/mol), while 25A75WT obtained the highest Ea (17.717 kJ/mol) among all the blends without a catalyst. For the blends with the catalyst, the activation energy was the lowest obtained (1.754 kJ/mol), which indicates that HZSM-5 could significantly reduce the activation energy.

## 4. Conclusions

The main objective of this work is to investigate the thermal kinetics and the synergistic effect of AP and WTS blends with and without the catalyst. A thermogravimetric analyzer was used to analyze the samples with different ratios and at three different heating rates. Results showed that the effect of heating rates throughout the process was negligible. In contrast, the co-feeding mixtures’ ratio significantly impacted the pyrolysis of the samples, thereby acting as a hydrocarbon donor, where 25A75WT had the highest rate loss, while 75A25WT and 75A25WT+C accelerated the reaction time by decreasing the initial and final temperature values. A promoting effect was dedicated in the blends 75A25WT+C and 75A25WT. The activation energy varies for all the blends, where the blend 25A75WT had the highest activation energy. In contrast, the blend 75A25WT+C had the lowest activation energy, which indicates that the presence of a catalyst can reduce the activation energy. The blend 75A25WT+C had the lowest activation energy.

## Figures and Tables

**Figure 1 ijerph-19-07101-f001:**
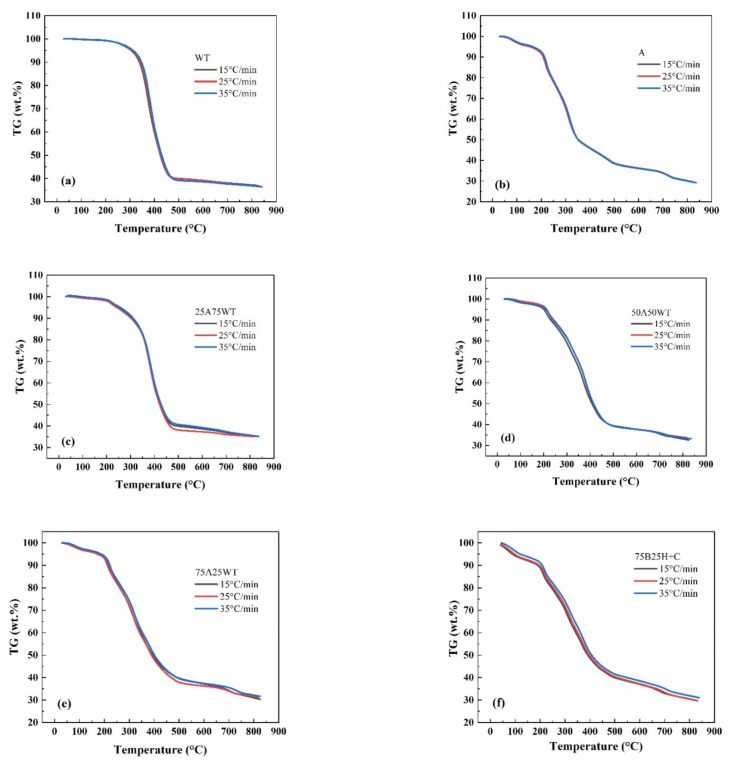
TG curves of (**a**) WTS, (**b**) AP, and (**c**–**f**) their blends at three different heating rates.

**Figure 2 ijerph-19-07101-f002:**
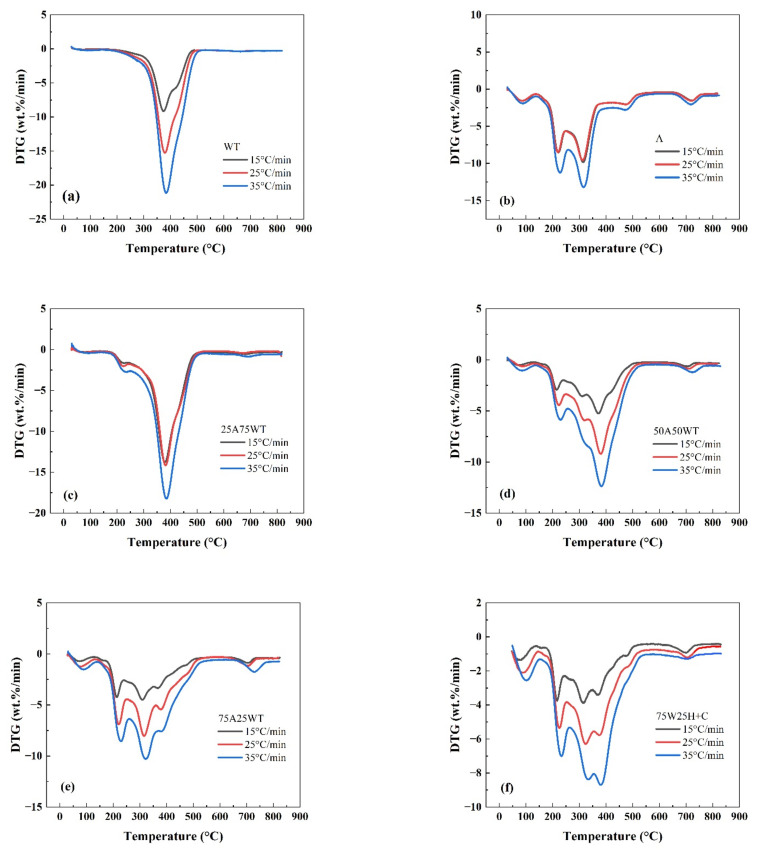
DTG curves of (**a**) WTS, (**b**) AP, and (**c**–**f**) their blends at three different heating rates.

**Figure 3 ijerph-19-07101-f003:**
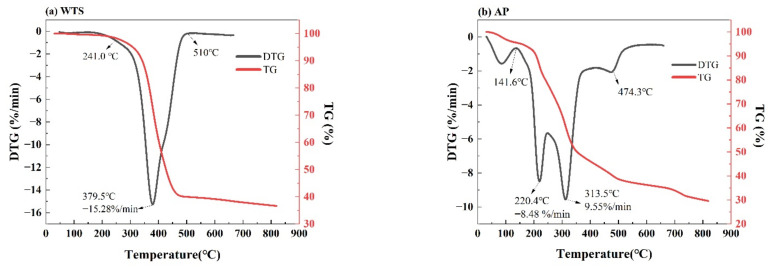
TG and DTG curves of (**a**) WTS and (**b**) AP at 25 °C/min.

**Figure 4 ijerph-19-07101-f004:**
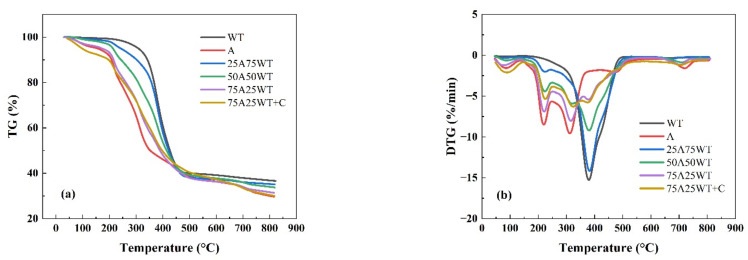
(**a**) TG and (**b**) DTG curves of AP, WTS, and their blends with and without HZSM-5 at 25 °C/min.

**Figure 5 ijerph-19-07101-f005:**
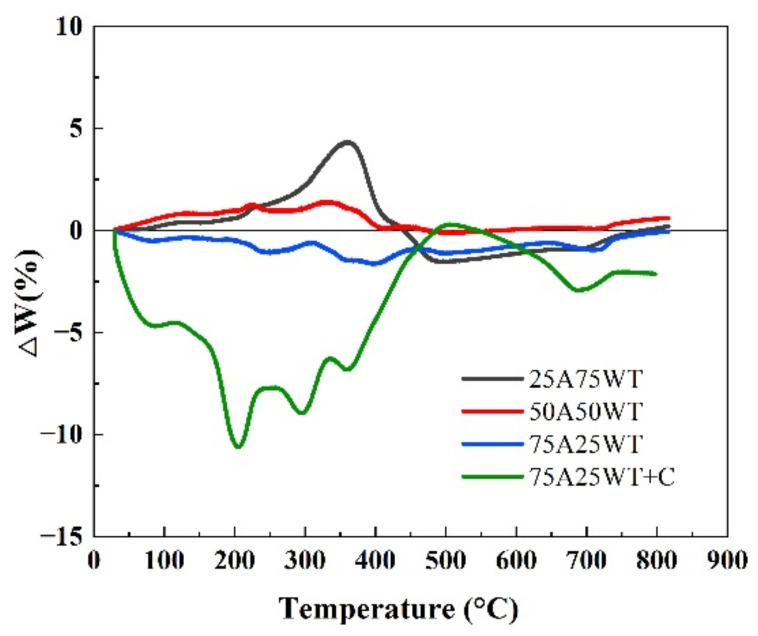
The fluctuation of ΔW for the blends at 25 °C/min.

**Figure 6 ijerph-19-07101-f006:**
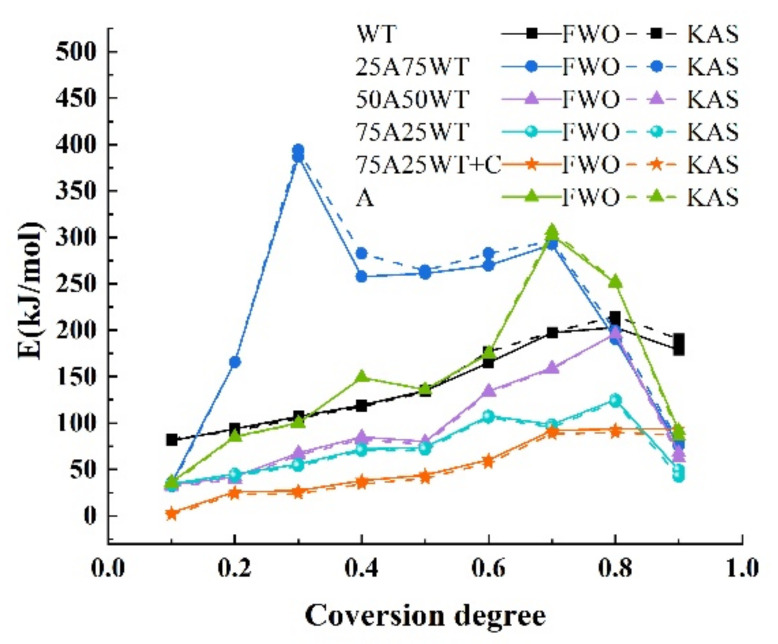
A comparison of activation energy and conversion degree distributions obtained by FWO and KSA for AP, WTS, and their blends.

**Figure 7 ijerph-19-07101-f007:**
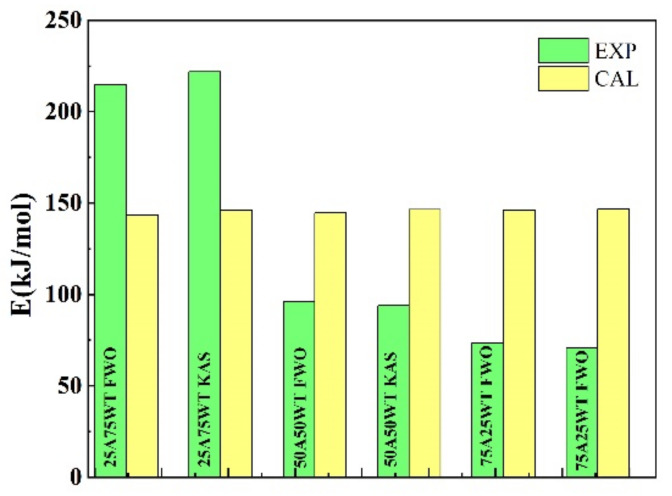
A comparison of the experimental and the calculated activation energy for the blends without HZSM-5.

**Figure 8 ijerph-19-07101-f008:**
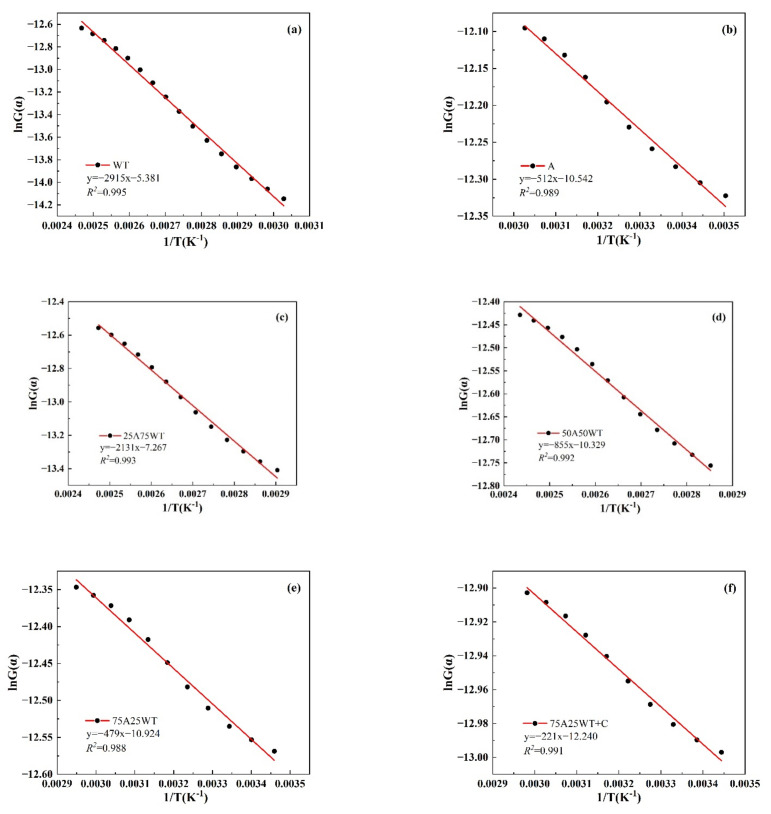
The slopes and intercepts of the samples of the plot ln(G(α)) vs.1/T for (**a**) WTS, (**b**) AP, and (**c**–**f**) their blends with and without HZSM-5.

**Table 1 ijerph-19-07101-t001:** Proximate and ultimate analysis of dried *A. philoxeroides*.

*A. philoxeroides*	
Proximate analysis:	
Moisture	4.58
Ash	17.35
Volatile	54.68
Fixed carbon	27.97
Composition analysis	
Hemicellulose	33.87
Lignin	25.68
Cellulose	27.51
Ultimate analysis (wt%):	
C	39.13
H	5.09
O	35.9
N	2.53

**Table 2 ijerph-19-07101-t002:** The main parameters of AP, WTS, and their blends with and without HZSM-5 obtained from the TGA at 25 °C/min.

Sample	1st Stage				2nd Stage				3rd Stage			
	$T_{s}\;({}^{\circ}C)$	$T_{f}\;({}^{\circ}C)$	$T_{max}\;({}^{\circ}C)$	$DTG_{max}$	$T_{s}\;({}^{\circ}C)$	$T_{f}\;({}^{\circ}C)$	$T_{max}\;({}^{\circ}C)$	$DTG_{max}$	$T_{s}\;({}^{\circ}C)$	$T_{f}\;({}^{\circ}C)$	$T_{max}\;({}^{\circ}C)$	$DTG_{max}$
WT	_	_	_	_	_	_	_	_	241.0	510.0	379.5	15.28
A	141.6	245.7	220.4	8.48	262.5	367.6	313.5	9.55	386.7	542.0	474.3	2.1
25A75WT	179.0	243.2	221.7	2	_	_	_	_	248.6	530.0	379.5	14.2
50A50WT	156.7	244.0	223.8	4.44	258.4	333.2	319.2	5.9	344.5	531.0	379.8	9.2
75A25WT	148.2	248.2	220.7	6.69	255.0	358.9	312.7	8	361.7	528.0	377.9	5.43
75A25WT+C	146.2	239.7	223.0	5.35	253.4	351.6	321.69	6.3	355.0	506.0	374.1	5.78

$T_{s}$: the temperature for starting mass loss; $T_{f}$: the temperature for final mass loss; $T_{max}$: the temperature for maximum mass loss rate; ${\mathrm{DTG}}_{max}$: the maximum mass loss rate.

**Table 3 ijerph-19-07101-t003:** Kinetic parameters obtained by FWO and KAS approaches at different conversion degrees.

		FWO		KAS				FWO		KAS	
	α	$E_{a}\;\left(kJ/mol\right)$	$R^{2}$	$E_{a}\;\left(kJ/mol\right)$	$R^{2}$		α	$E_{a}\;\left(kJ/mol\right)$	$R^{2}$	$E_{a}\;\left(kJ/mol\right)$	$R^{2}$
**WT**	0.1	80.9	0.955	82.8	0.961	**25A75WT**	0.1	35.2	0.985	32.0	0.969
	0.2	93.5	0.972	91.2	0.978		0.2	165.6	0.998	165.8	0.999
	0.3	106.9	0.954	105.0	0.964		0.3	386.8	0.990	394.2	0.996
	0.4	119.3	0.976	117.4	0.984		0.4	257.5	0.992	282.4	0.997
	0.5	134.8	0.969	133.4	0.978		0.5	261.0	0.952	264.3	0.964
	0.6	164.9	0.975	177.0	0.985		0.6	269.7	0.961	282.6	0.974
	0.7	197.2	0.971	197.6	0.980		0.7	292.3	0.968	296.0	0.978
	0.8	202.9	0.988	215.1	0.975		0.8	190.6	0.982	198.9	0.990
	0.9	178.7	0.985	190.5	0.970		0.9	75.5	0.934	79.6	0.966
**Average**		142.1		145.5		**Average**		214.9		221.7	
**50A50WT**	0.1	33.3	0.978	31.0	0.984	**75A25WT**	0.1	34.4	0.986	32.4	0.991
	0.2	42.0	0.999	39.2	0.995		0.2	45.0	0.987	42.8	0.993
	0.3	67.9	0.999	65.2	0.995		0.3	55.8	0.982	53.5	0.989
	0.4	85.0	0.981	82.2	0.966		0.4	71.9	0.951	69.7	0.960
	0.5	79.9	0.999	76.8	0.999		0.5	73.6	0.955	71.1	0.964
	0.6	134.3	0.990	132.8	0.996		0.6	107.6	0.964	105.8	0.973
	0.7	159.4	0.997	158.0	0.992		0.7	98.3	0.999	95.3	0.999
	0.8	196.0	0.978	196.0	0.986		0.8	125.1	0.968	122.7	0.976
	0.9	68.0	0.993	62.6	0.997		0.9	49.2	0.967	42.3	0.968
**Average**		96.2		93.8		**Average**		73.5		70.6	
**75A25WT**	0.1	3.3	0.982	1.2	0.970	**A**	0.1	36.5	0.954	34.3	0.964
**+C**	0.2	25.9	0.960	23.3	0.963		0.2	85.3	0.962	84.8	0.972
	0.3	27.1	0.994	23.9	0.998		0.3	100.1	0.977	99.6	0.985
	0.4	37.9	0.976	34.2	0.980		0.4	148.9	0.996	149.3	0.999
	0.5	43.8	0.967	40.2	0.972		0.5	136.1	0.976	135.9	0.985
	0.6	60.4	0.979	56.8	0.985		0.6	174.4	0.974	175.5	0.983
	0.7	91.7	0.995	88.3	0.988		0.7	302.3	0.976	307.2	0.985
	0.8	93.6	0.998	89.4	0.992		0.8	250.7	0.990	252.3	0.996
	0.9	93.6	0.975	87.1	0.981		0.9	91.8	0.999	85.7	0.999
**Average**		53.1		49.4		**Average**		147.4		147.2	

**Table 4 ijerph-19-07101-t004:** The calculated activation energy of the blends 25A75WT, 50A50WT, and 75A25WT.

	25A75WT	50A50WT	75A25WT
FWO	143.5	144.8	146.0
KAS	146.0	146.4	146.8

**Table 5 ijerph-19-07101-t005:** Activation energy and correlation coefficient obtained by CR approach.

	$E\left(kJ/mol\right)$	$R^{2}$
WT	24.235	0.995
A	4.257	0.989
25A75WT	17.717	0.993
50A50WT	7.108	0.992
75A25WT	3.982	0.988
75A25WT+C	1.754	0.991

## Data Availability

Not applicable.

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
