# Peer review of "Investigation of Synergistic Effects and Kinetics on Co-Pyrolysis of Alternanthera philoxeroides and Waste Tires"

_ijerph, 2022, doi:10.3390/ijerph19127101_

Round 1

Reviewer 1 Report

1. Page 4, what does the x in equation (1) represent? Equation (3) describes the error.

2. Page 4, line 152: reference format needs to be modified.

3. What do TsTfTmax and DTGmax in Table 2 represent respectively?

4. Page 10, line 325: 'increased from ? = 5 to ? = 7 '. The range of ? is 0 to 1, the ? in the paper is out of its range.

5. The unit of the abscissa in Figure 8 is K-1, the obtained temperature range is 285.71-416.67K, and the converted temperature range is 12.56-143.52°C, which is inconsistent with the experimental temperature range.

Reviewer 2 Report

The work is of particular interest to understanding to the synergistic effect of co-pyrolysis between waste (tyres) and biomass.

However, the reviewer is of the view that there are significant confusing, conflicting statements, and gaps in analysis that needs to be addressed.

line 17: Acronyms or labels in the abstract & conclusions (75A25WT, etc.) are recommended to be described in full.

line 47: inaccurate description 'can be used as biogas'

line 56: residence time of 2 seconds is too short to define a process as fast pyrolysis. The reviewer suggests that the authors to provide a range of reaction time from literature that performs fast pyrolysis. 

line 91: Poor grammar. "This study was conducted co-pyrolysis of..." 

line 112: Table 1, ash content of A.philoxeroides is 17%, which is significantly higher than lignocellulosic biomass. Moisture content for A.philoxeroides was not shown.

line 160: rephrase the slope 'of the lot the...' the reviewer is unsure of what the authors are describing. 

line 173-182: TG curves in Figure 1 for samples with mostly waste tyres did not exhibit a 4 stage weight loss. Please elaborate the reasons for this difference with A.philoxeroides.

line 192-198: the statement where a high heating rate is insuffient for surface & interior of the object simultaneously is confusing, because in all cases the weight loss at 35degC/min is consistently the highest, and there is no evidence where "there is insuffient heating for surface & interior". The choice for 25 degC/min thus becomes dubious. In addition, there are no significant differences in the TG curves in Fig. 1 with respect to the heating rate, but the weight loss or DTG curves is significantly different with respect to the heating rate in Fig. 2 by as much as 10 to 20%. I recommend the authors to analyze & review the data again for a better discussion of the results.

line 203: moisture evaporates at 100degC. How would AP start to decompose at 50degC due to moisture?

line 276: delta W for 50A50WT was negligible 'due to the biomass proportion' in the blend. This statement conflicts with earlier statements where WT inhibited the decomposition of biomass (line 236-237).

line 283: 25A75WT had a positive delta W up to 500degC. Again, another confusing statement with line 268-269. It should be made clear that at this composition, the amount of biomass did not have a promoting effect until temperatures higher than 500degC. 

line 300: statements should be revised.

Section 3.4.1: based on the TG & DTG curves, the activation energy for waste tyres should be the highest for all conversion degrees. Instead, 25A75WT had the highest activation energy. The discrepancy between the results require further analysis.

Table 3&4: too many significant figures/decimal points

Section 3.4.2: the CR model gave a better representation of the experimental results. However, a discussion in regards to the discrepancies in values between the activation energies from Section 3.4.1 and 3.4.2 is missing. Authors should elaborate on why the activations energies differ by a factor of almost ten in some cases. 

Reviewer 3 Report

Paper deals about investigations on thermal kinetics and synergistic effect between Alternanthera philoxeroides and waste tires. Investigations were conducted by using thermogravimetric analysis in a temperature range of 50-900°C by using different heating rate. Two different methods were used to calculate the activation energy of different blend among philoxeroides and waste tires. Experiences demonstrated that the addition of WTS improves the thermal degradation of the sample while no significant effects on activation energy were estimated varying heating rate. Best results in terms of lower activation energy were obtained with the 75A25WT blend. It was also clarified the effect of the addition of HZSM-5 in terms of reduction of the energy activation.

Introduction well emphasizes the question about Alternanthera philoxeroides and well discusses the synergistic effect between biomass and waste tyres. Furthermore, no indication about previous studies on A. philoxeroides thermal decomposition (i.e. pyrolysis, gasification, ..) was provided. This should inform the reader about the technical feasibility of Alternanthera philoxeroides to be gasified/pyrolized (very high water content of Alternanthera philoxeroides). It is my opinion that it is suitable to introduce more references (if available) about thermochemical treatment of Alternanthera philoxeroides.

Eq. 3 . For the reader it isn’t properly clear as Eq.3 was obtained. You defined beta=dT/dt, where T is the temperature. How do you get beta=da/dt à Eq.3 ? It is advisable to rewrite the equation in a more explicit manner.

A Nomenclature section must be provided.

Some typing errors are present. It is advisable to carefully read the paper. Some examples:

Page 2, row 73. Impede à Impedement? 

Page 9, row 275. however à However (h à H)

Page 14, row 398. … with and without the catalyst. A thermogravimetric …

It is my opinion that the paper could be accepted after minor revision.
